# Cannabinoid-Based Ocular Therapies and Formulations

**DOI:** 10.3390/pharmaceutics15041077

**Published:** 2023-03-27

**Authors:** Sofia M. Saraiva, Lucía Martín-Banderas, Matilde Durán-Lobato

**Affiliations:** 1CPIRN-IPG—Center of Potential and Innovation of Natural Resources, Polytechnic Institute of Guarda, Av. Dr. Francisco de Sá Carneiro, No. 50, 6300-559 Guarda, Portugal; 2Departamento Farmacia y Tecnología Farmacéutica, Facultad de Farmacia, Universidad de Sevilla, C/Prof. García González n °2, 41012 Sevilla, Spain; mduran@us.es; 3Instituto de Biomedicina de Sevilla (IBIS), Campus Hospital Universitario Virgen del Rocío, 41013 Sevilla, Spain

**Keywords:** cannabinoids, ocular, eye, drug delivery system, nanoparticles

## Abstract

The interest in the pharmacological applications of cannabinoids is largely increasing in a wide range of medical areas. Recently, research on its potential role in eye conditions, many of which are chronic and/or disabling and in need of new alternative treatments, has intensified. However, due to cannabinoids’ unfavorable physicochemical properties and adverse systemic effects, along with ocular biological barriers to local drug administration, drug delivery systems are needed. Hence, this review focused on the following: (i) identifying eye disease conditions potentially subject to treatment with cannabinoids and their pharmacological role, with emphasis on glaucoma, uveitis, diabetic retinopathy, keratitis and the prevention of *Pseudomonas aeruginosa* infections; (ii) reviewing the physicochemical properties of formulations that must be controlled and/or optimized for successful ocular administration; (iii) analyzing works evaluating cannabinoid-based formulations for ocular administration, with emphasis on results and limitations; and (iv) identifying alternative cannabinoid-based formulations that could potentially be useful for ocular administration strategies. Finally, an overview of the current advances and limitations in the field, the technological challenges to overcome and the prospective further developments, is provided.

## 1. Introduction

Marijuana, cannabis, hemp and weed are commonly used terms for *Cannabis sativa*, which has been used since ancient times (2700 a.c.) [1]. Due to its narcotic and psychotropic effects, it was consumed mainly as a recreational drug in early years, which undermined its consideration as a potential therapeutic [2,3]. However, its numerous pharmaceutical properties prompted further research on cannabis as a medicinal product, which proved to be prolific [2,4]. As a result, in the last years, different institutions have reviewed the pharmacological properties of cannabis, which has led to its decriminalization and availability for therapeutic use [5,6,7]. Some of the diseases for which cannabinoids have demonstrated therapeutic activity include neurological disorders, muscle spasticity, multiple sclerosis symptoms, spinal cord lesions and epilepsy, to mention a few [8]. Still, nowadays, the most well documented therapeutic benefit of cannabinoids, and therefore their wider prescription, is in the prophylaxis and management of nausea and sickness related to antineoplastic chemotherapeutics [9]. Nonetheless, several synthetic cannabinoids have entered the international pharmaceutical market, such as nabilone and dronabinol [10]. Some examples of FDA-approved products are Epidiolex^®^ (cannabidiol (CBD)) for the treatment of refractory epilepsy-associated seizures, Marinol^®^ and Syndros^®^ (Dronabinol, synthetic Δ9-THC), both for chemotherapy-induced nausea and vomiting and anorexia in patients with acquired immune deficiency syndrome (AIDS).

Among the increasing number of identified pharmacological properties of cannabinoids, a potential use in ocular diseases is also being considered. Even though many legislations specifically within the EU do not currently provide for the use of cannabis for the treatment of eye diseases [11,12], the current climate of renewed interest in cannabis medical use and legalization promotes research in this area. Many of these conditions are chronic and even incapacitating and currently do not come with a convenient or sufficiently effective treatment, hence the need for alternative therapies [13]. The therapeutic potential of cannabinoids in ocular pathologies is mainly due to their role in the endocannabinoid system. This physiological system is constituted by several receptors, mainly cannabinoid receptors CB_1_ and CB_2_ (Figure 1), and their natural endogenous ligands, such as anandamide (N-arachidonoylethanolamine, AEA) and 2-arachidonoylglycerol (2-AG), which are synthesized on demand by different enzymatic pathways [14] (Figure 2). In addition, non-cannabinoid receptors such as transient receptor potential vanilloid 1 (TRPV1), G-protein-coupled receptor 18 (GPR18), GPR55, GPR119 and peroxisome proliferator-activated receptors (PPARs), GABA receptors and ion channels are also activated by endocannabinoids (Figure 1) [15]. Thus, their use is being considered for an increasing number of ocular syndromes [16].

However, cannabinoids generally present unfavorable physicochemical and biological properties, such as poor stability, hydrophobicity and an oily resin nature, among others [17,18]. These hurdles overall limit their bioavailability and access to the therapeutic target.

Thus, drug delivery tools are required to enable them to exert an adequate pharmacological effect. In this arena, nanotechnology-based approaches are gaining increasing relevance [14,19,20]. 

The main advantage nanotechnology can provide relies on the possibility of delivering virtually all types of active molecules in a localized fashion. To that aim, the active pharmaceutical ingredient (API) of interest is loaded or associated with a specifically designed nanocarrier, which subsequently improves its solubility and stability against degradation, while modulating its interaction with the cells or tissues of interest [21,22]. Thus, drug bioavailability and the subsequent therapeutic outcome are markedly improved. A variety of biocompatible nanostructured formulations has been developed to better fit the therapeutic agents of interest and the targeted site of delivery, such as nanoparticles (NPs), nanoemulsions (NEs), micelles, liposomes and lipid nanoparticles, among others [21]. Specifically in the field of ocular delivery, nanosystems were initially proposed in the 1980s [21,23]. Later on, several strategies were explored aimed at improving the capacity of nanocarriers to improve their ocular residence time and permeation and deliver drugs to the ocular anterior and posterior tissues, including the modulation of their physicochemical properties (i.e., size, surface charge) and the inclusion of mucoadhesive compounds, targeting moieties or PEGylation [24,25]. Altogether, these approaches were proven to be especially successful, ultimately leading to improved drug efficacy and decreased reduced toxicity [21,24,25].

Overall, cannabinoids present promising and highly desirable properties for the treatment of ocular pathologies, but this research topic is still in its infancy. As such, despite the well demonstrated usefulness of nanotechnology for ocular drug delivery, its use in ocular cannabinoid delivery is underexplored.

In this sense, this review has the purpose of analyzing which cannabinoid applications have been identified for ocular pathologies and other diseases as well as which nanotechnology-based strategies have been and could be implemented to design novel formulations to optimize the therapeutic potential of cannabinoids. 

## 2. Ocular Drug Delivery

The treatment of ocular conditions presents a challenge for ophthalmologists due to ocular anatomy and physiology. This small organ is composed by different static and dynamic barriers that make it difficult for the delivery of drugs to the target ocular structure. In the same way, the secondary effects caused by the available marketed products highlight the need to investigate new treatments [25]. To improve drug efficacy and reduce side effects, ocular therapeutics can be administered by different routes. 

The selection of the most suitable route of ocular administration depends mainly on the localization of the ocular target structure or cells, namely the anterior or the posterior eye segments. Topical instillation (eye drops) is usually the most used type of administration of ocular therapeutics for the treatment of conditions affecting the eye’s anterior segment, for its simplicity and for allowing the patient to self-administer the therapeutic. However, despite the exposure and accessibility of the eye, as previously mentioned, the anterior segment presents several barriers. The tear film composition (lipids, aqueous fluid, mucins and enzymes), its constant turnover and the blinking reflex are the first barriers encountered and are responsible for the fast clearance of topically administered drugs. As such, only about 5% of the applied dose is maintained in this segment [24,26].

The drugs that can surpass these barriers and reach the cornea must overcome the different epithelial, stromal and endothelial layers that form this tissue. In addition, both the cornea and conjunctiva present tight junctions, which further limit drug permeation. Furthermore, if the drug targets the inner tissues of the anterior segment, it also needs to overcome the aqueous humor, which contains proteins and low molecular weight compounds that may interfere with the process [24,26].

On the other hand, when the target tissues are located in the posterior segment (choroid, retina, optical nerve), topically, administered drugs will have to overcome the aforementioned barriers and then diffuse through the dense matrix of the vitreous humor composed of collagen, hyaluronic acid and proteoglycans before reaching the tissues in the back of the eye. Considering the severity and impact of the chronic vitreoretinal diseases that affect this segment, such as macular degeneration, retinitis pigmentosa and diabetic retinopathy, among others, and the difficulty to reach this segment by topical administration (lack of suitable permeation), the management of these diseases usually requires intravitreal (IVT) and subretinal injections [25]. 

Oral administration of drugs for the treatment of ocular conditions can also be used; however, only about 1–2% reaches the eye [27]. This route is highly attractive and convenient for patients, and therefore is a highly common route of administration. However, it is associated with drug-related systemic side effects. Furthermore, when the target tissue is the eye, orally administered drugs have to overcome the different biological barriers and challenges such as the gastrointestinal tract barriers (harsh pH environments, mucus, degrading enzymes), bloodstream (protein opsonization) [28] and ocular posterior segment barriers (blood–retinal barrier (BRB)) [27]. Likewise, other systemic routes (intravenous (IV) or intramuscular injections) must overcome the bloodstream and ocular barriers as well, in addition to being painful. 

Aside from the challenges inherent to the route of administration that can impact the success of a therapeutic, drugs’ physicochemical properties such as hydrophobicity and instability, characteristic of cannabinoids and other molecules, pose an extra challenge [17,29]. These parameters should be carefully addressed to enable a safe administration and improve residence time and permeation through ocular tissues. Therefore, in the last decades, relevant efforts have been dedicated to the development of novel and cost-effective formulations for ocular application in order to improve therapeutic efficacy [29,30]. 

In this scenario, nanostructured systems have been gaining relevance, either alone or in combination with traditional systems, in the form of drops for topical instillation in the eye, mainly pursuing an improved release [31]. Their capacity to protect labile drugs against degradation and overcome ocular barriers makes them an especially valuable tool for ocular drug delivery in general and for the ocular delivery of challenging molecules (i.e., biomacromolecules, poorly water-soluble molecules) in particular [32,33]. These novel systems must comply with the general requisites of conventional pharmaceutical products, such as sterility, isotonicity, stability and safety [34]. Nevertheless, the performance of these advanced drug release systems is determined by a series of additional properties that must also be addressed and properly controlled [29].

### Properties of Ocular Drug Delivery Nanosystems

Particle nanosystems are characterized as being colloidal systems whose size oscillates between 10 and 1000 nm. Through the control of a series of factors, nanocarriers can present different functionalities and applications, which would not be feasible using traditional formulations. 

One of the requirements that nanosystems must comply with is being based on compositions that, aside from ensuring efficacy, are safe. Proteins, lipids, as well as polymers derived from natural and synthetic sources, namely chitosan, albumin, sodium alginate, polylactic acid (PLA), poly-lactic-co-glycolic acid (PLGA) or polycaprolactone (PCL), lecithin, phosphatidylcholine, oleic acid, vitamin E and medium-chain triglycerides (MCT), among others, have been used [27,35,36].

An especially relevant parameter is the nanovehicle **size**, due to its influence on tissue penetration and capacity to maintain a controlled drug release, thereby maximizing the dosing period [31]. The control of this parameter would improve the patient’s quality of life and reduce the associated costs of medical visits due to the frequent injections, usually necessary to complete the treatment [25].

Another parameter that influences the penetration capacity of nanosystems is their **surface**. The superficial properties of nanosystems can increase the residence time of the formulation. As previously described, one of the reasons for the low availability of drugs upon their topical administration is their low retention caused by their elimination through the constant renewal of the lacrimal fluid, nasolacrimal drainage and degradation of the drug [34]. Nanoparticulate systems allow for an increased residence time at the tissue due to their surface functional groups as well as their superficial charge, which enable the interaction of the particles with mucins, thereby prolonging the presence of the drug in the cornea. Moreover, in order to increase the residence time of NPs in the precorneal compartment, some compounds such as chitosan and hyaluronic acid (HA) or polyethylene glycol (PEG) have been used [24,31]. Among them, chitosan is the most widely used for such purposes, since it presents a positive net charge that is attracted to the negative surface charge of the cornea, contributing to prolonging the permanence in the nanostructure of the cornea and decreasing its clearance [29]. 

Another advantage that nanosystems offer derives from their capacity to encapsulate drugs, which protects them from enzymatic degradation. As a consequence, the dose of drug necessary to achieve the desired effect is lower [34]. This feature is especially dependent on the **composition and structure** of the nanocarrier, which in turn is closely related to all of its physicochemical attributes. In this regard, different types of nanosystems have been proposed for cannabinoid delivery, some of which were specifically designed for ocular cannabinoid delivery. They are mainly lipid- and polymer-based systems, being the first type the most studied for this purpose, as depicted in the present review. While each formulation must always be considered independently, and specific works will be discussed in the following sections, several common features of each of these types of nanosystems in terms of their composition and structure are worth mentioning.

Among the lipid-based nanosystems, nanosized oil droplets (oleic acid, medium-chain triglycerides (MCT), soybean oil, sesame oil, vitamin E) stabilized by surfactants and amphiphilic lipids (polysorbate 80, polysorbate 20, Span 20, Poloxamer 188, Pluronic F68, lecithin, phosphatidylcholine), known as NEs, have been shown to improve retention time and corneal penetration through tight junctions thanks to a careful selection of their composition, thereby enhancing the bioavailability of the drugs in the ocular tissues [37,38,39]. In addition, NEs are easily scaled-up and low-cost formulations. Currently, there are several commercially available formulations of NEs for ocular conditions, such as Cationorm^®^, Lipimix™, Cycloket^®^, Restasis^®^ and Ikervis^®^ [24,36].

On the other hand, solid lipid nanoparticles (SLNs) are formed by a solid lipid core (stearic acid, Compritol 888 ATO, Gelucire^®^ 44/14) stabilized by surfactants (Pluronic F-68, polysorbate 80, poloxamer 182) [40]. The solid core of these nanosystems has been reported to protect drugs from degradation and to be able encapsulate both hydrophobic and hydrophilic drugs [41,42].

In addition, liposomes are one of the most extensively studied systems for drug delivery through a wide range of administration routes. These nanosystems are mainly composed of a phospholipid bilayer including cholesterol and an aqueous core [43]. Despite the different advantages that these systems present (ability to encapsulate both lipophilic and hydrophilic drugs and biodegradability and biocompatibility, among others), liposomes are also prone to drug leakage and aggregation [36,41].

Finally, regarding polymer-based nanosystems, polymeric NPs and micelles have also been proposed for cannabinoid delivery to ocular tissues. Polymeric NPs, as the name indicates, are nanosystems composed of natural and synthetic polymeric networks, such as chitosan, hyaluronic acid, albumin, poly (lactic-co-glycolic acid) (PLGA), poly(lactic acid) PLA and poly(oxide ethylene) (PEO) NPs [24,44]. These systems are characterized by their biocompatibility and biodegradability, as well as their ability to prolong residence times. However, a burst release effect is commonly observed. On the other hand, polymeric micelles are formed by the self-assembly of amphiphilic polymers (e.g., chitosan/poly(vinyl alcohol) (PVA) grafted with poly(methyl methacrylate) blocks [45]), which leads to the formation of spherical structures with a lipophilic core and a hydrophilic corona. They usually present a smaller size than NPs, which can enhance their permeability through tissues, and are easily scaled up with a low cost of production [30,45].

Overall, through the modulation of the mentioned factors, including the selection of the best fitting type of nanocarrier, the drug concentration reaching the target site can be increased and maintained through prolonged periods of time. In the following sections, specific cannabinoid-based formulations, with a special focus on nanocarrier-based strategies aiming at solving the challenges of several ocular pathologies, will be discussed.

## 3. Vehiculated Cannabinoids for Ocular Pathologies

As previously mentioned, the available information on cannabinoid vehiculation for ocular applications is currently still scarce. Nonetheless, the therapeutic potential of these compounds in combination with their unfavorable physicochemical properties, as well as the ocular barriers against the penetration of drugs, are motivating research into this area for a wide range of ocular pathologies. 

### 3.1. Glaucoma

Glaucoma is, after cataracts, the second leading cause of blindness worldwide, which is a serious and extremely impactful consequence that could be avoided in 95% of cases with an early diagnosis [46]. Glaucoma involves a group of multifactorial optical neuropathies that are associated with a progressive loss of retinal ganglion cells which leads to vision loss [47].

The increase in intraocular pressure (IOP) is the major risk factor for the onset and progression of glaucoma, even though it is not the only one. IOP, which usually oscillates between 10 and 20 mmHg, is determined by an equilibrium between the production of aqueous humor by the ciliary body and its elimination through the pores of the trabecular meshwork. Therefore, increases in IOP can be due to the following: (i) an increase in aqueous humor production; (ii) iridocorneal angle closure not allowing the aqueous humor to reach the trabecular meshwork; or (iii) blockage of the trabecular meshwork, which hinders the outflow of the aqueous humor towards the circulatory torrent. Furthermore, other factors related to the toxic damage of ocular cells have also been reported to be related to glaucoma generation [48]. Therefore, glaucoma can be classified depending on three main variables, namely the opening of the iridocorneal angle, the origin and the moment of onset. Regarding the opening of the iridocorneal angle, glaucoma can be classified as open-angle, also known as chronic simple glaucoma, a chronic disease due to the increase in IOP affecting the optical nerve and leading to vision loss. On the other hand, closed-angle glaucoma is a pathology in which there is a sudden increase in ocular pressure and requires immediate treatment. Considering the origin, glaucoma can be classified as primary or secondary, being the last one produced by identified causes such as certain drugs. Lastly, according to the onset moment, glaucoma can be congenital, infantile, juvenile or have a later onset at an adult age [49,50].

Chronic simple glaucoma cannot be prevented to date. Plus, most patients do not present any symptoms, which highlights the importance of an early diagnosis. The diagnosis must be performed by a specialist by means of different tests, such as the measurement of ocular pressure and the determination of lateral and peripheral vision [48,51].

Currently, glaucoma is considered a chronic condition for which treatment is focused on delaying its progression and, if possible, arresting it, through the decrease in IOP levels, ultimately reducing its negative impact on the quality of vision of patients. The selection of the therapeutic depends on the type of glaucoma and the very patient, even though in the majority of cases, treatment is based on the topical administration of β-blockers, sympathomimetic drugs, prostaglandin analogs, carbonic anhydrase inhibitors or a combination of these [52]. Eye drops decrease IOP by reducing the production of aqueous humor or by improving its drainage. 

However, some patients do not respond to the available therapeutics and the condition aggravates. In such cases, surgery is necessary to facilitate the drainage of the aqueous humor, either by establishing a drainage fistula for the aqueous humor towards the exterior or by inserting an implant to enable the drainage [53,54]. Furthermore, the available therapeutics can also lead to adverse side effects such as hypotension, bronchospasm and iris pigmentation, among others. In the same way, the surgical option also presents disadvantages such as increasing the risk of cataracts [55,56]. In this sense, novel therapeutic approaches are needed. 

#### 3.1.1. Therapeutic Potential of Cannabinoids on Glaucoma and Limitations of Conventional Formulations

Cannabis seems to be an alternative for the treatment of glaucoma, since some studies have demonstrated its capacity to decrease IOP [16]. Back in the 1970s, the observation of the ocular hypotensive effect of cannabis in a group of healthy volunteers [57] and glaucoma patients [58,59] that smoked cannabis led to an increase in the research on cannabinoids for glaucoma.

This effect was attributed to Δ9-THC, which interacts with the CB_1_ receptors of the cannabinoid system, stimulating G_i/o_ proteins and subsequently inhibiting adenylyl cyclase, thereby hampering the conversion of adenosine triphosphate (ATP) into cyclic adenosine monophosphate (cAMP) [60]. These cAMP molecules can also regulate the activation of potassium channels, leading to a decrease in aqueous humor production and an increase in its elimination and consequently to a reduction in IOP levels [61,62]. Aside from CB1 receptors, GPR18 and GPR119 receptors have also been proven to regulate ocular pressure and to be activated by THC. In addition to the effect of Δ9-THC on IOP level management, this cannabinoid was also reported to reduce the loss of retinal ganglion cells [63], which highly impacts disease progression and subsequent vision loss. Similar effects were also observed using other synthetic cannabinoids, such as HU-211 injected intravitreally [64], topically administered WIN55212-2 [65,66] and VSN16S [67] and BW146Y administered orally using capsules [68].

Despite the capacity of Δ9-THC to reduce IOP levels, the effects of CBD on IOP are contradictory. According to some studies, the cannabinoid CBD caused an increase in IOP after sublingual administration (40 mg of CBD, oromucosal spray) in glaucoma patients [69,70] and topical instillation (5 mM; 0.16% *w*/*v* in Tocrisolve® vehicle (1:4 ratio soybean oil/water emulsified with Pluronic F68)) in mice [71]. Miller et al. also showed that, aside from its IOP-increasing effect, CBD also prevented the THC IOP-decreasing effect, raising concerns about its use by patients suffering ocular hypertension with glaucoma risk. Nonetheless, the study by Tomida et al. showed that this effect was transient and dose-dependent, occurring only with the 40 mg dose of CBD but not with a lower dose (20 mg). In addition, other works have shown no effects of CBD on IOP levels after IV and oral administration in animal models [72,73,74]. Furthermore, other studies showed that CBD caused a reduction in IOP levels upon topical administration in animal models either dissolved in mineral oil [75] or vehiculized in NEs [38,76].

Regarding side effects, it should be noted that both the local and systemic adverse effects of cannabinoids are mainly related to the route of administration. In the case of Δ9-THC, side effects such as increases in cardiac function, drops in cardiac pressure, cold sweats and paleness and anxiety, among others, were observed upon systemic administration, namely by inhalation, ingestion or IV administration [58,59,77,78,79]. Therefore, providing a localized application that could minimize such adverse effects would be of major interest. When targeting ocular administration, the highly lipophilic character of cannabinoids should be carefully taken into consideration. Δ9-THC has an aqueous solubility of 1–2 µg/mL and a logP of 6.42, which makes it very poorly soluble in tear fluid [40] and leads to the need to use solubilizing formulations that can adequately dissolve these drugs. 

Initial attempts to administer cannabinoids topically to the eye were limited to the use of mineral oil [72,80] (Table 1). According to a study by Green et al., a single topical administration of Δ9-THC in mineral oil (concentration not specified) caused minor ocular irritation (burning sensation and tearing) and did not reduce IOP levels. Similar effects were observed after one week of treatment (Δ9-THC 1% in mineral oil, four times/day) [80]. More specifically, no change in IOP levels was observed and several patients, mainly those receiving the vehicle alone (mineral oil), abandoned the study due to adverse side effects (burning sensation and lid swelling). 

Other vehicles, such as polysorbate 80 (surfactant), ethanol and dimethyl sulfoxide (DMSO) were also considered for the same purpose [78]. Polysorbate 80 and ethanol are only tolerated by the eye within a limited range of concentrations and have been approved by the U.S. Food and Drug Administration (FDA) as inactive ingredients for ophthalmic use in different dosage forms. Specifically, polysorbate 80 is approved for ophthalmic application in the form of solution/drops and emulsions up to 0.15% and 4% *w*/*v*, respectively, while ethanol in solution form is approved for this application up to 0.5 % *w*/*v*. On the other hand, the use of DMSO for ophthalmological use has only been approved by the European Medicines Agency (EMA) as an idoxuridine cosolvent (Antizona, UK). However, these formulations have presented limitations in the forms of cytotoxicity or irritation [81]. In addition, the use of cyclodextrin (CD) inclusion complexes as a solubilizing approach was also explored in this context. Specifically, the topical administration of the CD-complexed CB1 receptor agonist WIN55212-2 led to limited but promising effects on glaucoma patients [66].

Therefore, the use of advanced drug delivery systems to optimize the administration and effects of cannabinoids would provide significant advantages. Due to its lipophilicity, cannabinoids are more soluble in oily carriers than aqueous ones. In addition, these carriers can provide additional benefits for drug absorption. In this sense, several studies have demonstrated cannabinoids’ effects upon topical delivery using a commercial soybean oil emulsion stabilized by Pluronic-F68 and Tocrisolve^®^ [40,71,82]. For instance, Miller et al. used Tocrisolve^®^ to study the Δ9-THC and CBD effects on topical administration in mice [71]. A single administration of Δ9-THC (5 mM) caused a 28% reduction in IOP levels in male mice which persisted for at least 8 h. The effect of Δ9-THC was shown to be sex-dependent, causing a reduction in IOP in females only after 4 h, which returned to base levels in less than 8 h post-treatment. The highest effect observed in males was attributed to the higher expression of CB_1_ and GPR18 receptors in males. Other researchers used Tocrisolve^®^ for the delivery of a Δ9-THC prodrug (Δ9-THC-valine-hemisuccinate, THC-VHS) [40], CBD prodrug [83] and synthetic cannabinoid HU308 [84], among others. Despite being useful for studying cannabinoid effects, Tocrisolve^®^ may not be a suitable solution for enhancing cannabinoid delivery and its therapeutic effect. For instance, Adelli et al. [82] showed that after 60 min of Δ9-THC-Tocrisolve^®^ topical administration, Δ9-THC was found in the iris ciliary (IC) bodies (53 ng/50 mg) and choroid retina (5 ng/50 mg), while THC-Val-HS was present in the aqueous humor (9 ng/100 µL), IC bodies (24 ng/50 mg) and choroid retina (15 ng/50 mg). In addition, the prodrug (THC-VHS)-Tocrisolve^®^ (equivalent to 0.6% *w*/*v* Δ9-THC) showed an IOP-modulating effect similar to the one obtained with the marketed pilocarpine (2% *w*/*v*) and timolol (0.25% *w*/*v*) drops but with a shorter duration than timolol (2 vs. 6 h, for the prodrug and timolol, respectively). THC-VHS has a higher stability, hydrophilicity and solubility than Δ9-THC, provided by amide linkages, enhancing its penetration capacity and IOP-lowering effects [39,40,82]. However, a more efficient carrier is needed to prolong its effect and provide a greater therapeutic effect than the currently available drugs.

**Table 1 pharmaceutics-15-01077-t001:** Recent cannabinoid formulations designed for topical ophthalmic administration for the management of ocular diseases.

Targeted Disease	Cannabinoid	Formulation	Main Outcome	Reference
Glaucoma	Δ9-THC	Mineral oil	No reduction in IOP levels, side effects (burning sensation, tearing, lid swelling)	[72,80]
		Tocrisolve^®^ (commercial soybean oil emulsion (1:4 oil-in-water) stabilized by Pluronic-F68)	Sex-dependent IOP decrease; 28% IOP peak reduction persisting at least 8 h in male mice, persisting 4 h in female mice	[71]
	THC-VHS, Δ9-THC	Tocrisolve^®^ (commercial soybean oil emulsion (1:4 oil-in-water) stabilized by Pluronic-F68)	36% IOP peak reduction for 2 h with Δ9-THC; 47% IOP peak reduction for 4 h with THC-VHS, resulting in IOP-modulating effect similar to commercial formulations but with shorter duration (2 h vs. 6 h)	[82]
	Δ8-THC	NEs (soybean oil, oleic acid, phospholipids, poloxamer, α-tocopherol, glycerin)	Marked and prolonged (over 8 h) decrease in IOP in normo- and hypertensive rabbits	[85]
	THC-VHS	NEs (sesame oil, polysorbate 80, poloxamer^®^188), alone and combined with a mucoadhesive agent (Carbopol^®^)	Higher and longer prolonged decrease in IOP vs. standard commercial treatments in New Zealand rabbits; high concentrations of polysorbate 80 led to diminished effect, attributed to THC entrapment in surfactant micelles decreasing permeation and release	[39]
		SLNs (Compritol 888 ATO, Pluronic F-68, polysorbate 80, glycerin)	Improved residence time and bioavailability; IOP peak decrease (31%) with longer prolonged effect (480 min) than standard commercial treatments (120 and 180 min of pilocarpine and timolol maleate, respectively)	[40]
	CBD	Tocrisolve^®^ (commercial soybean oil emulsion (1:4 oil-in-water) stabilized by Pluronic-F68)	IOP increase, prevention of THC IOP-decreasing effect	[71]
		NEs (sesame oil, polysorbate 80, poloxamer^®^188), alone and combined with a mucoadhesive agent (Carbopol^®^)	IOP peak reduction (19.9%) maintained for up to 300 min in normotensive rabbits	[38]
	WIN55212-1	45% *w*/*v* 2-hydroxylpropyl-β-cyclodextrin in pH 7.4 adjusted saline	31% IOP peak reduction within 1 h after single administration, maintained for 2 h	[66]
	CBGA	PEO/PLA NPs in an in situ gelling hyaluronic acid (HA) and methylcellulose (MC) hydrogel	300-fold higher corneal penetration ex vivo in porcine whole eyes vs. CBGA in mineral oil, accounting for 0.015% of applied CBGA	[44]
Keratitis	Δ8-THC, CBD and HU-308	Soybean oil	Reduced pain scores, neutrophil infiltration and inflammation	[86]
	CB1 allosteric ligand GAT211 and enantiomers GAT228 and GAT229, Δ8-THC	Soybean oil with 2% DMSO and 4% Tween 20	Reduced pain scores (GAT228), reduced corneal inflammation (GAT228 and GAT228 with Δ8-THC	[87]
	CBD	Mucoadhesive micelles of chitosan/poly(vinyl alcohol) and poly(methyl methacrylate)	Permeation through human cell corneal epithelium monolayer in vitro, up to 86% and 53% of applied CBD reaching the acceptor compartment in liquid–liquid and air–liquid exposition, respectively	[45]
		NEs (medium-chain triglycerides (MCT), polysorbate 80 and Solutol^®^ HS 15), antioxidants propyl gallate or butylhydroxytoluene	Decrease in key inflammatory cytokines and IOP, not following a standard sigmoidal dose–response	[76]
Uveitis	HU308	Tocrisolve^®^ (commercial soybean oil emulsion (1:4 oil-in-water) stabilized by Pluronic-F68)	Reduced leukocyte levels in the iris microvasculature, decreased proinflammatory mediators, higher anti-inflammatory effects vs. reference compounds (nepafenac, dexamethasone, predinosolone)	[84,88]
	RO6871304, RO6871085, HU910	Tocrisolve^®^ (commercial soybean oil emulsion (1:4 oil-in-water) stabilized by Pluronic-F68)	Attenuated leukocyte adhesion to the iris microvasculature	[89]
Dry Eye Syndrome	Δ9-THC	15% *w*/*v* DMSO and 10% *w*/*v* Cremophore EL in saline	Protected corneal nerve morphology, maintained corneal sensitivity, reduced infiltration of inflammatory CD4+ T cells	[90]
	Δ8-THC, CBD, and HU-308	Soybean oil	Antinociceptive and anti-inflammatory effects	[86,87]

#### 3.1.2. Recent Advances of Nanosystems in Enhancing Therapeutic Effects of Cannabinoids on Glaucoma

In light of this evidence, nanocarriers and especially lipid-based nanostructures, represent a promising alternative. In this context, cannabinoid-loaded NEs were first proposed for cannabinoid topical ocular administration by Muchtar et al. [85] (Table 1). The instillation of Δ8-THC-loaded NEs in ocular hypertensive rabbits led to a significant and prolonged IOP decrease without causing irritation [85]. Recently, Sweeney et al. [39] evaluated the capacity of NEs, composed of sesame oil, polysorbate 80 and poloxamer^®^188 (oil, surfactant and cosurfactant), in the ocular delivery of THC-VHS to manage IOP (Table 1). The effect of polysorbate 80 concentration on THC-VHS-loaded NEs’ physicochemical properties and the IOP effect when topically instilled in New Zealand rabbits was studied. A surfactant concentration of up to 2% *w*/*v* improved the decrease in IOP and prolonged its effect for about 360 min; nonetheless, at 4% *w*/*v* (polysorbate 80), the effect was shortened to 90 min. The authors hypothesized that the highest surfactant concentration might have caused the entrapment of THC in micelles formed within the NEs, thereby causing a decrease in corneal permeation and drug release [39]. In addition, the formulations containing 2% *w*/*v* of polysorbate 80 were also able to cause a higher decrease in IOP levels than the control THC-VHS-Tocrisolve^®^ and the standard treatments latanoprost and timolol [39], leading to a maximum drop in IOP of 23, 21, 13 and 14%, respectively. Furthermore, THC-VHS-NE (1% *w*/*v*, equivalent to 0.6% *w*/*v* THC) led to a more prolonged IOP decrease (at least 480 min) than THC-VHS-Tocrisolve^®^ (240 min). Moreover, the authors studied the possibility of including a mucoadhesive compound to increase the viscosity of the NE formulation, aiming at improving its ocular residence time and thus the duration of the therapeutic effect. Carbopol^®^ 940NF, a polymer approved by the FDA for ophthalmic products, was selected for its solubility in acidic conditions, in which THC-VHS is highly stable. In addition, upon contact with tear fluid, the pH increases and the polymer forms a viscoelastic gel [91]. This inclusion of Carbopol^®^ 940NF as a mucoadhesive agent caused an increase in the formulation viscosity and NE particle size from about 96 to 160 nm without compromising the monodisperse profile of the formulation. Overall, the resulting final formulation showed increased ocular residence time, without hindering drug diffusion and release, which finally resulted in a greater IOP decrease and prolonged effect than THC-VHS-NEs without Carbopol^®^ or the commercial latanoprost eye drops (0.005% *w*/*v*) as well [91].

Considering the promising results obtained, more recently, the same group developed a similar mucoadhesive NE formulation for the encapsulation of CBD (1% *w*/*v*) [38] (Table 1). As previously performed for THC-VHS, CBD was solubilized in sesame oil (NE oily core). As previously observed by Sweeney et al. [91], the addition of Carbopol^®^ caused not only an increase in NE particle size, but also an increase of surface charge, namely from 167 to 260 nm and −20 to −38 mV (NEs compared to Carbopol-NEs), attributed to the coating of the NEs with the polymer. Despite the contradictory effects of CBD on IOP levels found in the literature, Senapati et al. found that CBD-NEs caused a significant reduction (19.9%) in the IOP levels of normotensive rabbits, maintaining this effect for at least 300 min. 

Other recent works have explored the delivery of cannabinoids using lipid-based NPs for the treatment of glaucoma by means of topical administration. SLNs, prepared by ultrasonication using Compritol 888 ATO, Pluronic F-68, polysorbate 80 and glycerin, were developed to improve the ocular administration of THC-VHS (Table 1). Compritol was selected for its low melting point (70 °C) and high capacity to solubilize THC-VHS, and Pluronic F-68 and polysorbate 80 act as stabilizers of the nanosystem. The SLNs increased the residence time of the prodrug in the eye in rabbits, enhancing their bioavailability in the choroid retina, in comparison to THC-VHS-Tocrisolve^®^ [40]. These results were attributed to the selection of SLNs based on their composition and their capacity to interact with the ocular mucosa, which improve their residence time and drug bioavailability [92,93]. Furthermore, Taskar et al. showed that a single dose (50 µL) of THC-VHS-SLNs (0.6% THC equivalent) accomplished a marked decrease in IOP (31%), with a more prolonged effect (during 480 min) than the commercial solutions of 2.5% (*w*/*v*) pilocarpine hydrochloride (120 min) and 0.25% (*w*/*v*) of timolol maleate (180 min) [40]. 

Other lipidic nanosystems, namely liposomes, were also evaluated for THC delivery for IOP management, even though the intended administration was systemic in this case. Specifically, Szcesniak et al. administered THC-loaded liposomes through the intratracheal and intraperitoneal (IP) routes and evaluated the effect on IOP [43]. A faster reduction in IOP was obtained with a lower dose of the cannabinoid with intratracheal vs. IP administration. However, due to the rapid and unspecific distribution of the liposomes, the duration of the effects was short [43]. Again, a local ocular administration would represent a promising approach in this scenario.

Finally, a combined formulation of NPs with hydrogels as a way of improving ocular residence time was also explored with the cannabinoid cannabigerolic acid (CBGA) in this context. Specifically, Kabiri et al. produced CBGA-loaded PEO/PLA NPs. CBGA was used as a model molecule without pharmacological activity, due to its similar pharmacokinetic profile to other cannabinoids approved for ocular management (Table 1). In addition, the authors also developed a hydrogel composed of HA and methylcellulose (MC), which presented a sol–gel transition at 31.5 °C. Therefore, the hydrogel comprising NPs had the capacity to be formed in situ upon contact with the ocular surface, increasing the contact time of the nanosystems with the ocular surface. The evaluation of the formulation performance, through ex vivo studies performed using whole porcine eyes, showed a 300% improvement in corneal drug penetration in comparison to the control group (CBGA in mineral oil). Despite the improvement, only 0.015% of the initial amount of applied CGBA permeated through the cornea, which was attributed to the lack of lachrymal fluid drainage [44].

### 3.2. Keratitis

Keratitis is an inflammation that affects the cornea. This pathology can be generated by multiple causes, with bacterial or viral infections being some of the most frequent. Other potential causes are ocular dryness, allergies and amoeba-induced infection, among others [94,95]. Keratitis usually produces an intense ocular pain, tearing, photosensitivity and eye redness due to the congestion of the blood vessels that surround the cornea [95]. Contact lens users and those patients who have suffered a trauma on the surface of the cornea are more prone to develop infectious keratitis [96]. The current treatment depends on the origin of the keratitis. In the case of infectious keratitis, the treatment is selected depending on the origin of the infection, while in the case of non-infectious keratitis, a palliative treatment is applied.

The modulation of the endocannabinoid system has become a focus for the treatment of pain and inflammation. Recent studies have demonstrated that the activation of the CB_1_ receptor in the cornea by the topical administration of Δ8-THC decreased the corneal pain induced by the activation of the transient receptor potential vanilloid type 1 channel (TRPV1) and also reduced the infiltration of neutrophiles [86,87]. In addition, the activation of this receptor was also shown to reduce the expression of proinflammatory mediators after a corneal lesion (Figure 3) [97]. Therefore, the available results overall demonstrate that the CB_1_ receptor plays an important role in corneal wound healing [97]. Hence, the activation of this receptor represents an interesting target to control neuropathic corneal pain through the modulation of pain sensation and the inflammatory response with subsequent sensitization induced over time. Once again, a local method of action on the eye receptors would be an interesting approach to minimize possible systemic side effects and potentiate the efficacy of cannabinoids. 

Following this strategy, Sosnik et al. developed CBD-loaded micelles of chitosan/PVA) and poly(methyl methacrylate) for corneal delivery, aimed at ocular conditions involving inflammation such as keratitis, uveitis and others (Table 1) [45]. The mucoadhesive micelles presented adequate physicochemical properties regarding size (100–200 nm), positive surface charge (+32 to +38 mV) and adequate loading capacity (20%). In addition, in vitro assays performed in a human corneal epithelium model demonstrated the capacity of these CBD micelles to permeate through corneal cell monolayers under liquid–liquid (LL) and air–liquid (AL) conditions. AL conditions differentiate from the LL setup by exposing the cells to air and showing the formation of stronger tight junctions between the cells, which more closely mimics the conditions of the corneal cells in the eye [45]. Transepithelial electrical resistance (TEER) and fluorescence measurements demonstrated the capacity of the nanosystems to permeate through the cell layers in both conditions. The results were further corroborated by the quantification of 86% and 53% of the administered CBD encapsulated in the NEs, after 4 h of treatment, in the acceptor compartment of the cell inserts cultured under LL and AL conditions, respectively, proving the capacity of the micelles to overcome biological barriers and deliver the drug. 

Another recent work, performed by Benita’s group, evaluated the potential of CBD-NEs for the treatment of keratitis using a *Pseudomonas aeruginosa* LPS keratitis murine model [76]. The CBD-NEs were composed of medium-chain triglycerides (MCT), polysorbate 80 and Solutol^®^ HS 15, and presented a hydrodynamic diameter and surface charge of 148 nm and −29 mV, respectively (Table 1). The NEs’ pH (6.3) and isotonicity (290–300 mOsm) were adjusted by the addition of sodium hydroxide and glycerin. Finally, the inclusion of the antioxidants propyl gallate or butylhydroxytoluene in the formulation was necessary to maintain CBD stability up to three months. Regarding the performance of the NEs, the topical instillation of the formulation (2 μL) with a CBD concentration above 0.4% *w*/*v* led to a decrease in the levels of key inflammatory cytokines involved in corneal damage. In addition, the authors reported that the treatment with CBD-NEs did not follow a standard sigmoidal dose–response. Particularly, at the 0.4 and 1.6% *w*/*v* CBD concentrations, IOP levels were reduced, but at the 0.8% *w*/*v* concentration, no alteration was observed [76]. This was attributed to several potential causes, namely the following: (i) the action of CBD in other cannabinoid receptors aside from CB_1_, such as GPR18 and GPR19, which regulated IOP through different mechanisms; (ii) the presence of CB_1_ in both sites of the aqueous humor (inflow and outflow), triggering different mechanisms [98]; and (iii) the experimental conditions set in the study, such as performing the study during the animal dark cycle, among other factors.

### 3.3. Uveitis

Uveitis is a heterogeneous group of clinical syndromes with multiple causes that have in common the inflammation of the uveal tract and/or its adjacent structures. This condition can be a manifestation of an established generalized disease, a process limited to the ocular globe or the first sign of a pathology that will be developed over time [99,100]. The current prevalence of this pathology in developed countries is 5.4 per 1000 subjects and its onset usually falls within the fourth decade of a patient’s life, even though it can be developed at any age [100,101]. Different genetic, immunological and environmental factors are involved in the development of this condition [100,101], which hampers its diagnosis. In addition, regarding diagnosis, which is initially clinical [102], there is no common agreement on the laboratory tests or images that should be acquired for a correct detection in and monitorization of patients. 

In terms of treatment, the most common therapeutic option is the use of corticosteroids either by systemic or ocular (topical or intraocular) routes [103]. Nonetheless, the long-term use of corticosteroids is commonly associated with adverse side effects, such as the increase in IOP levels and the susceptibility to infections, myopia and subconjunctival hemorrhage [103], which calls for the identification of novel therapeutic anti-inflammatory agents. For instance, new therapies have been developed based on immunomodulation and immunosuppression, and great benefits are expected from these novel agents, such as adalimumab (Humira^®^) [104]. However, this approach is associated with several important drawbacks, such as being expensive, requiring subcutaneous injections once every two weeks [105,106] and the risk of causing important side effects such as infections (tuberculosis, pneumonia), nervous system problems, heart failure and immune reactions, among others [107]. Therefore, efficient and cost-effective therapies with less side effects are of great interest. 

In this scenario, cannabinoids have drawn attention due to the reported role of CB_2_ receptors in ocular inflammatory and immune reactions [108,109]. Specifically, the activation of CB_2_ receptors by JWH133 (3-(1′1′-dimethylbutyl)-1-deoxy-Δ8-THC) was reported to exert anti-inflammatory effects in the retina in an autoimmune uveoretinitis (EAU) mouse model [110]. The synthetic cannabinoid JWH133 was dissolved in a mixture of ethanol and polysorbate 80, the ethanol was removed before reconstitution in phosphate-buffered saline (PBS) and the resulting preparation was administered through IP injection. The treatment resulted in the inhibition of the infiltration of inflammatory cells in the retina, cytokine/chemokine production and T cell activation [110]. 

In another study, the topical administration of HU308 (1.5 % in Tocrisolve^®^) caused a reduction in the leukocytes in the iris microvasculature and a consequent decrease in ocular proinflammatory mediators, thereby ameliorating the pathology of the disease (Figure 4). In addition, the administration of HU308 led to higher anti-inflammatory effects than other commonly used anti-inflammatory compounds such as Cox inhibitors (nepafenac) and corticosteroids (dexamethasone and prednisolone) (Figure 5) [88]. Later, the same research group showed that the effect of HU308 in the reduction in leukocyte infiltration in a uveitis mouse model was caused by a non-CB_2_ target and that the reduction in neutrophil migration was mediated trough CB_2_ receptors, as well as the regulation of lipid signaling pathways (prostaglandins, lipoamine 2-acyl glycerols) which contribute to the regulation of inflammation [84].

More recently, Porter et al. demonstrated that the topical application of synthetic cannabinoid agonists of CB_2_ RO6871304 and RO6871085, as well as HU910 (1.5% *w*/*v* in Tocrisolve^®^), was effective to attenuate the adhesion of leukocytes to the iris microvasculature, especially for RO6871304, in uveitis induced by endotoxins [89]. 

Overall, the available studies indicate that the targeting of CB_2_ receptors and the development of selective ligands are promising anti-inflammatory approaches to treating uveitis, proliferative vitreoretinopathy [111] and other ocular inflammatory diseases [110]. Considering these effects achieved by different routes of administration of CB_2_ agonists, the vehiculation of cannabinoids, especially through the use of nanocarriers for topical administration, would again represent a promising therapeutic approach.

### 3.4. Dry Eye Syndrome

Dry eye syndrome (DED) is characterized by a deficiency of tear fluid that hinders proper eye lubrication. DED can be caused by a deficient production of the tear fluid (volume and/or composition) or its increased evaporation. This is usually associated with discomfort, redness, burning and itching sensation, blurred vision, eye pain and photophobia, overall impacting the patient’s quality of life. In advanced stages, it can lead to infection, inflammation and damage of the corneal tissue [112,113,114]. This condition can be aggravated with age, lifestyle, environmental conditions, as well as requiring refractive surgery and causing chronic blepharitis, and is also commonly associated with other autoimmune diseases [112,113]. Currently, the available therapeutic options consist of artificial tear substitutes, gels and anti-inflammatories. Some examples of the currently marketed products are Restasis^®^, Ikervis^®^ and Cyclokat^®^ (cyclosporine NEs), Cequa^®^ (micelles), and Cationorm^®^ and Lipimix™ (drug-free NEs) [24]. However, the available treatments are only able to relieve symptoms (dryness, pain, burning sensation) and inflammation [112,113]. Therefore, novel approaches that could improve the management of this condition are necessary. 

Cannabinoids, due to their capacity to modulate ocular pain, inflammation and wound healing [70,86,87], may be a relevant therapeutic approach for DED. The ocular pain associated with this condition is mainly derived from the damage to the corneal epithelium and inflammation, which triggers the release of nerve growth factor (NGF), which consequently activates the TRPV1 receptor [113]. This non-cannabinoid receptor, found in the cornea and conjunctiva, seems to colocalize with CB_1_ receptors. CB_1_ and CB_2_ are expressed in the corneal epithelium and endothelium, as well as in the conjunctival vascular endothelium and stromal cells [15]. However, CB_1_ was also found expressed in the axons of cholinergic neurons innervating the lacrimal gland [70]. The expression of these receptors, which is usually low, increases under stress, such as DED-associated inflammation. McDowell et al. showed that cannabinoid analgesic effects upon CB_1_ activation could be due to the subsequent reduction in NGF-induced sensitization of TRPV1 in sensory nerves [115]. Recently, Tran et al. studied the effect of Δ9-THC as a non-selective CB_1_ and CB_2_ agonist along with SR141716A and SR144528, CB_1_ and CB_2_ selective antagonists, respectively, in a DED mouse model. Topically administered Δ9-THC, formulated in a mixture of 15% *w*/*v* DMSO and 10% *w*/*v* Cremophore EL in saline, was able to protect corneal nerve morphology and maintain corneal sensitivity, as well as reduce the infiltration of inflammatory CD4+ T cells, by activating CB_1_ and CB_2_ receptors, respectively [90]. Similarly, Thapa et al. showed that topically applied Δ8-THC dissolved in soybean oil (0.2–5.0% *w*/*v*) exerted antinociceptive and anti-inflammatory effects, which were mediated by CB1 (Figure 6) [86,87]. However, CBD and HU-308 (CBD derivative), dissolved and applied in similar manner, showed anti-inflammatory and antinociceptive effects mediated by the activation of the receptors 5-HT1A and CB_2_, respectively [86].

On the other hand, the effect of cannabinoids on tear film was also considered. To the best of our knowledge, only one study focused on cannabinoid effects on tearing [70]. The activation of neuronal CB_1_ receptors in the lacrimal gland seemed to influence tear film production. Furthermore, the effect of IP administration of THC on tearing was dependent on gender, specifically male mice, which presented a higher expression of CB_1_ receptors than female mice, and suffered a reduction in tear production while in females, tearing was not affected. In addition, the CB_1_ receptor agonist (CP55940 at 0.5 mg/kg) also decreased tearing in male mice, but increased it in female mice, after IP injection [70]. Considering the impact of eye lubrication on DED, the role of cannabinoids in this context should be much further studied.

Overall, the mentioned studies provide evidence on the beneficial role of cannabinoids for the treatment of corneal pain and inflammation and therefore support conducting further research on their potential for the management of DED and other inflammatory conditions affecting ocular surface tissues, such as keratitis and uveitis.

### 3.5. Diabetic Retinopathy

Diabetic retinopathy (DR) is characterized by the decomposition of the blood–retinal barrier (BRB) and neurotoxicity, which have been associated with proinflammatory cytokines and oxidative stress [116]. This progressive and asymptomatic disease is the result of a vascular damage, which is characterized by an increase in permeability and capillary vessel damage [117]. DR is one of the leading causes of preventable blindness in the adult working population and the fifth leading cause of blindness and moderate and severe vision impairment in patients 50 years old and older [118]. The risk factors are very diverse, among them the evolution period of the diabetes and the type of diabetes. Nonetheless, the control of arterial hypertension, hyperlipemia and anemia caused by diabetes decrease the onset and progression of DR. Likewise, there are associated hormonal changes, such as puberty, that can accelerate the onset of DR. In addition, several genetic factors have been identified to be involved. Finally, from an ocular perspective, there are different factors that can be associated with a higher risk of developing DR, including ocular hypertension, inflammation, and ocular trauma. In this sense, there are other factors that can protect against DR such as myopia [117,118,119]. 

Two main types of retinopathies can be differentiated: non-proliferative DR and proliferative DR [118]. Regarding treatment, the most successful alternative currently available is prevention. Several studies have demonstrated the extreme importance of glycemic control under treatment in the prevention and stabilization of DR [117,120]. From an ophthalmological point of view, prevention is based on early detection and adequate monitorization [117,118]. In the cases where prevention is not sufficiently effective, laser photocoagulation is the current standard treatment for DR [120]. Nevertheless, laser photocoagulation burns and destroys part of the retina and, consequently, central vision can be lightly impaired, night vision can be reduced and focus capacity can also be diminished. In addition, some patients can also lose part of their peripheral vision. Despite the consequences that can derive from laser photocoagulation therapy, the associated vision loss is low when compared to the vision loss resultant from untreated DR. In this context, the identification of alternative treatments could provide significant improvements. 

Cannabinoids have also been postulated as promising therapeutic tools for DR owing to their anti-inflammatory and antioxidant properties. Different studies have shown that cannabinoids reduce oxidative stress and inflammation by providing a neuroprotective effect [116,121]. A study developed by El-Remessy et al. [116] showed that the induction of neuronal cell death of the inner retina and the breakdown of the BRB during the first stages of diabetes in rats was correlated with an increase in the production of reactive oxygen species (ROS) and proinflammatory cytokines, with the activation of kinase MAP-p38. Treatment with CBD (IP injection, 10 mg/kg, every 2 days) reduced the production of ROS and hindered the activation of kinase MAP-p38. On the other hand, this study also revealed that CBD treatment evaded two functional components of DR: (i) vascular permeability and (ii) neuronal cell death. Therefore, the researchers demonstrated that the neuroprotective and conservative effects of CBD on the BRB were related to its antioxidant and anti-inflammatory properties (Figure 7) [116]. Considering that the administration of CBD was performed through IP injection, local administration optimized by means of an adequate formulation could much further improve treatment outcome.

### 3.6. Ocular Damage Caused by Pseudomonas aeruginosa Biofilms in Contact Lenses

Biofilms are an assembly of microorganisms surrounded by an extracellular matrix synthesized by themselves. Bacteria can form biofilms on natural surfaces, cardiac valves and inert structures such as contact lenses [122], being involved in most infectious processes. Their capacity to generate biofilms on the surface of contact lenses, the difficulty in eliminating them once formed and their infectious potential on the cornea upon microtraumas make biofilms an important risk factor in the development of bacterial keratitis in contact lens users [122,123]. 

*Pseudomonas aeruginosa* is one of the microorganisms with the capacity to form this type of structure. It can be found in humid environments, but also in the microbial flora of the humid areas of the skin [124,125]. Transmission usually occurs through the contact of injured skin or mucosa with contaminated water or objects. Thus, this microorganism can cause ocular infections, which in this case, are mainly associated with the contamination of the liquid used to clean the contact lenses. The use of contaminated ocular products can generate keratitis which can in turn lead to corneal melting and perforation, scar infection and even loss of vision in the affected eye [124,125]. 

Considering the devastating effects of this microorganism on ocular health, the prevention of biofilm formation and their elimination in contact lenses are essential to prevent infections and subsequent complications in contact lens users. Therefore, it is essential that contact lens solutions have the capacity to reduce the amount of pathogens and avoid or prevent biofilm formation on the lenses [126]. To that purpose, current strategies include coatings with anti-biofilm agents and the development of therapies based on them [127]. For instance, El-Galniny et al. evaluated the use of natural compounds, namely Calendula officinalis and Buddleja salviifolia extracts, in inhibiting and eliminating bacterial biofilms on soft contact lenses, which proved to be able to prevent and destroy them [126]. In line with this, different studies highlighted the antimicrobial and anti-inflammatory properties of Cannabis sativa, proposing its use in the development of therapies and coatings based on its metabolites [128].

Specifically, one study evaluated the capacity of CBD oil extract to inhibit and eliminate a *Pseudomonas aeruginosa* biofilm on soft contact lenses and compared the results with a multiuse commercial solution and a natural solution of fermented Allium sativum (BGE) [129]. The minimum inhibitory concentration of CBD (2% *w*/*v*) hindered the formation of biofilms in about 70%, both in standard clinical strains and in *Pseudomonas aeruginosa* strains isolated from eye swabs. This capacity of CBD can be considered highly effective considering that the BGE at 4% (*w*/*v*) and multiuse solutions (50% *v*/*v* of the initial concentration) showed lower inhibitory capacities (55% and 50%, respectively) [129]. Regarding its capacity to eliminate biofilms formed by the eye-extracted strains, CBD showed a 24% efficacy in both strains, which was however lower than the ones obtained with BGE and the multiuse solution, 36% and 41%, respectively. Furthermore, the authors also studied the efficacy of the combination of the multiuse solution with CBD. This combination resulted in a synergetic anti-biofilm effect of 75%, while the combination of CBD with BGE solution was slightly lower (69%) [129]. In sum, CBD metabolites displayed an important effect on the inhibition of biofilm formation and a moderate effect on the elimination of preformed biofilms, as well as the capacity to potentiate the anti-biofilm effect of a standard commercial multiuse solution [129]. Thus, CBD offers an interesting approach as an antimicrobial in the ophthalmological field. Again, the use of formulations in this area could provide additional advantages, such as controlled cannabinoid release for a prolonged and sustained effect.

## 4. Other Cannabinoid Advanced Vehiculation Strategies and Considerations for Ocular Applications

Considering the pharmacological potential of cannabinoids in the treatment of ocular diseases is a recent focus of research interest, and the available information on advanced drug delivery systems for their local administration to the eye is, consequently, very limited and recent, but expected to increase in the coming years. Thus, to the best of our knowledge, there are still no clinical data available on cannabinoid ocular delivery by means of nanoformulations. For instance, the results so far reported in this context refer mostly to systemic cannabinoid administration [16], along with few attempts at ocular topical administration using mineral oil as a vehicle [16,72,80] or CD inclusion complexes [16,66], previously commented on in this review. Overall, cannabinoid systemic administration led to inconsistent efficacy and/or side effects, while the use of mineral oil as a topical vehicle did not provide significant pharmacological effects and did lead to side effects, while modest but promising effects were obtained through the use of CD complexes. As commented on in previous sections, these observations could be ultimately explained by the unfavorable physicochemical and biological properties of cannabinoids along with restrictive ocular barriers, where the use of nanocarriers could be a valuable approach. It is also worth mentioning that the interest in cannabinoid delivery is also increasing greatly for a wide range of other therapeutic applications [14] and hence, the knowledge generated on cannabinoid encapsulation is expected to be translated to several administration strategies. Potentially, those cannabinoid-based formulations not initially developed for ocular administration could be considered for this purpose, providing they are compatible with the requisites of ophthalmic administration. Despite the unfavorable physicochemical properties of cannabinoids, successful cannabinoid encapsulation with adequate physicochemical properties of the resulting nanocarrier was achieved with polymeric NPs [130,131,132]; SLNs [133,134,135]; nanostructured lipid carriers [136]; NEs [137]; micelles [138,139,140]; microemulsions [141]; ethosomes [142]; pickering emulsions [143]; silica NPs [144]; and carbon nanotubes [145], to mention a few.

Overall, the most promising alternatives involving cannabinoid drug delivery in general, and ocular cannabinoid delivery in particular, are those displaying biocompatibility, high cannabinoid encapsulation efficiency and sustained release. Additional desirable features are their capacity to modulate particle size, surface properties and scalability. Lipid-based and polymeric nanocarriers specially comply with these requisites, although polymeric NPs tend to present a slower release of about days, in comparison to lipidic NPs that release the drug within hours [133]. Overall, the selection of the most adequate nanosystem would be subjected to the pathology of interest, the intended ocular administration route, the required dose and the physicochemical behavior of the final formulation.

Even though there are already studies reporting the application of these types of NPs for ocular administration, there are still limitations to be solved. Therefore, the performance of these nanosystems is susceptible to further improvement by optimizing the previously described factors (Section 3.3), such as the surface functionalization of NPs and conjugation of specific ligands of target cells [34]. Finally, the performance of nanosystems and the influence of each design modification should be evaluated through several relevant assays, some of which seem to be frequently obviated such as drug release and colloidal stability in simulated lacrimal fluid, compliance with the specifications for ocular formulations such as isotonicity and sterility, interaction in cell cultures, tissue biodistribution and in vivo efficacy, among others.

## 5. Conclusions and Future Perspectives

To date, there is evidence for the potential of cannabinoids in the management or treatment of several diseases. Regarding ocular applications, these drugs have shown interesting pharmacological properties that could lead to alternative treatments for several diseases in need of new or improved therapies, including glaucoma, uveitis, DR, keratitis and the prevention of *Pseudomonas aeruginosa* infection. However, many of the mechanisms of action involved are still not fully elucidated, and the complexity of the endocannabinoid system along with the multiple potential interactions of cannabinoids hinder this endeavor. In addition, cannabinoids also present important side effects, especially when administered through a systemic route. Finally, owing to the different barriers and protection mechanisms of the eye, the development of therapeutics for this organ is highly challenging. 

In line with this, along with further research on the role of cannabinoids in ocular pathologies, the development of delivery formulations for local administration to the eye could provide a relevant breakthrough in this context. Specifically, nanotechnology tools, including the selection of materials and methodologies as well as technological adjustments for administration through different routes, are showing great potential to overcome the challenges of both cannabinoid vehiculation and ocular topical delivery, ultimately aiming at targeted delivery.

Currently, there is a limited, though recent, number of studies on the development and evaluation of cannabinoid-based formulations for the ocular route. Relatively simple designs of nanosystems have presented several advantages for ocular topical administration, including increased permeation and residence time and lower administration frequency. Nonetheless, in order to reach the ophthalmological market, increasing the amount of drug that reaches the site of action is still a required optimization. To that aim, a series of parameters should be controlled, such as size, surface properties, encapsulation, release capacity and biocompatibility, among others. In a further step, advanced designs including targeting moieties or responsive nanosystems could also be considered.

The optimization of these systems could be also useful for the treatment of other pathologies and the knowledge gathered from cannabinoid delivery systems already developed for other pathologies and administration routes could also provide valuable information in this context. For instance, several nanocarriers have been assayed as cannabinoid delivery systems, including polymeric NPs, lipid-based systems (SLN, NLC, liposomes, micelles, NEs, microemulsions) and carbon nanotubes, among others. Based on these efforts, nanostructures composed of hydrophobic materials seem to provide the best option for cannabinoid vehiculation due to cannabinoids’ lipophilic character, while polymeric materials also offer the possibility of amenable chemical modification for surface tuning and targeting. 

Altogether, the pharmacological application of cannabinoids in ocular diseases is a new and growing research field with great potential, albeit facing several challenges that will require multidisciplinary efforts from the scientific community and governments. Looking ahead, extensive research is expected to be conducted on the mechanistic effects of single cannabinoid molecules as well as with their combinations, in consideration of the cannabinoid entourage effect. In line with this, analytical and technical advances related to the robust production of cannabis extracts and derivative products with standardized contents are also expected to promote the development of medical applications of cannabinoids. Subsequently, from the side of governments, the generated knowledge on their mechanisms of action and necessary guidelines ensuring their safe use will potentially facilitate the relaxation of legislation regulating its medicinal use. In addition, and as mentioned, the actual translation of cannabinoid pharmacological potential into therapies is expected to require advanced drug delivery tools to overcome their physicochemical and biological limitations. In line with this, further evolved drug delivery systems are expected to be proposed for this purpose. Specifically, special consideration should be given to the selection of a specific composition for the selected drug to be delivered and its final target, leading to robust and controlled physicochemical attributes of the resulting nanocarriers, including size and surface properties, drug release and colloidal stability in biologically relevant fluids. Additionally, further particle surface modification by covalent linkages of targeting moieties is a feature expected to be incorporated as well. Finally, additional aspects such as nanocarrier biocompatibility, product storage stability and the possibility to scale up the production of these nanocarriers will need to be considered in order to allow for an actual clinical translation. Overall, the development and optimization of cannabinoid-based delivery systems for ocular therapy offers a fruitful area for the generation of valuable new treatments with a long way ahead.

## Figures and Tables

**Figure 1 pharmaceutics-15-01077-f001:**
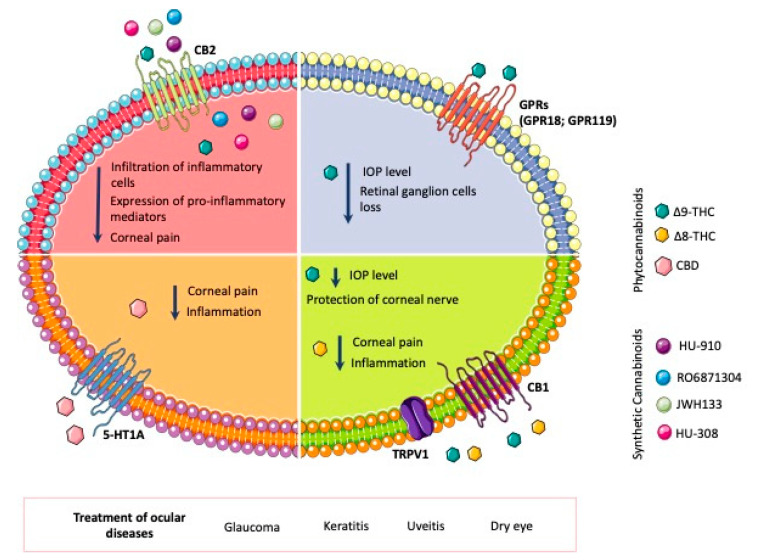
Cannabinoid signaling pathways and effects. Regardless of the type of cannabinoid ligand (phytocannabinoid, endocannabinoid or synthetic), these compounds primarily interact with G protein-coupled receptors (GPCR), such as the CB1 and CB2 receptors and GPR55, or with transient receptor potential (TRP) channels, such as TRPV1, to induce a cellular response. The activated pathways vary based upon receptor activation and have physiological effects on pain, inflammation and intraocular pressure levels, among other effects.

**Figure 2 pharmaceutics-15-01077-f002:**
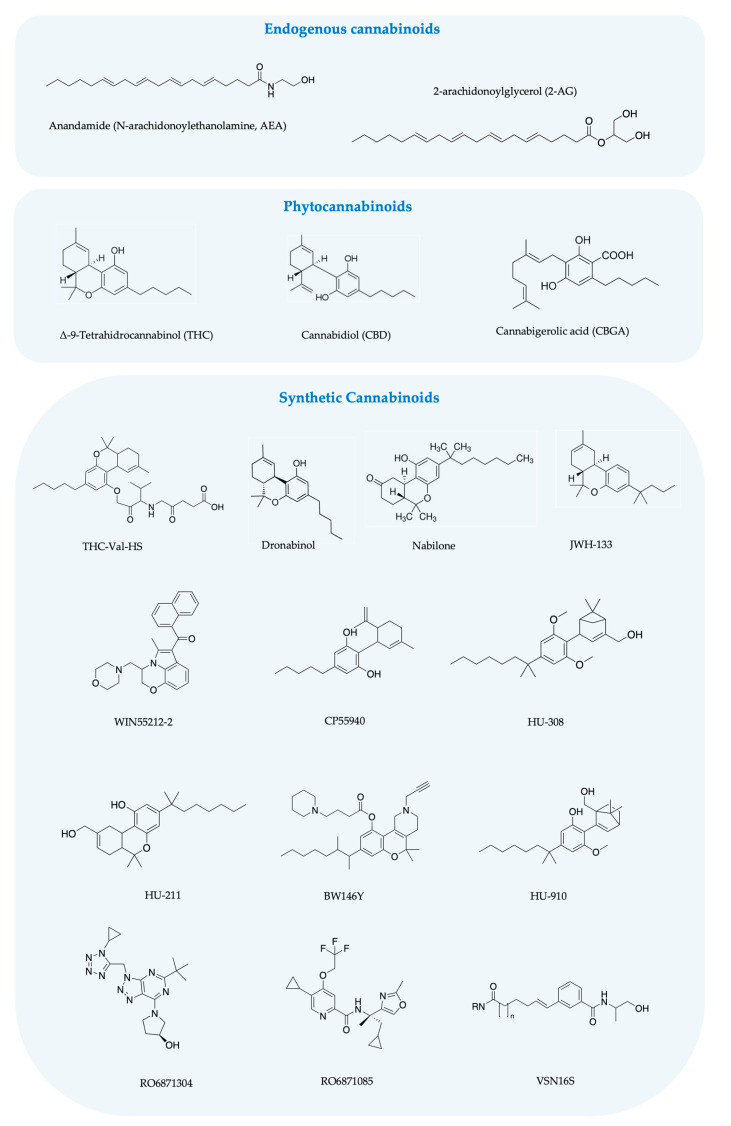
Chemical structures of endocannabinoids, phytocannabinoids and synthetic cannabinoids with reported pharmacological potential in ocular diseases.

**Figure 3 pharmaceutics-15-01077-f003:**
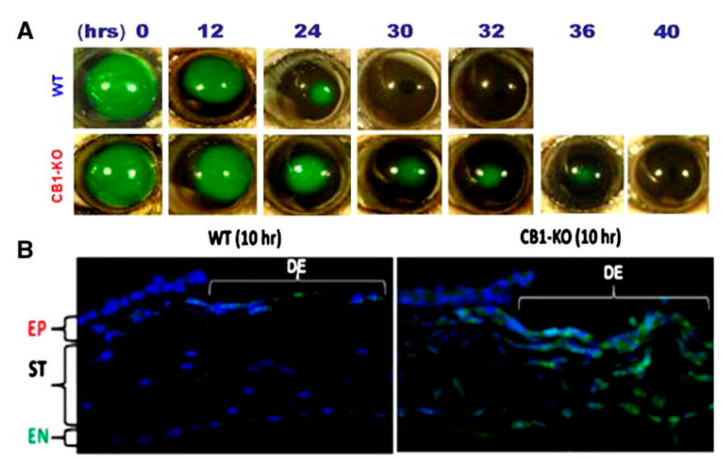
CB1 activation promoted mouse corneal reepithelialization. (**A**) Time-dependent epithelial wound healing was evaluated following epithelial debridement with an Algerbrush in control (WT) and CB1−/−mice. The wound area was identified based on the extent of fluorescein staining. Corneal reepithelialization in the WT group required 30 h, whereas it was delayed by 10 h in the CB1−/− group since closure occurred at 40 h. (**B**) Immune cell infiltration was assessed at 10 h in WT (left panel) and CB1−/− (right panel). CD 11b green immunostaining in the remaining deepithelialized area (DE) was only evident in the CB1−/− (KO) cornea. Scale bar = 50 μm. Reproduced from Yang et al. [97].

**Figure 4 pharmaceutics-15-01077-f004:**
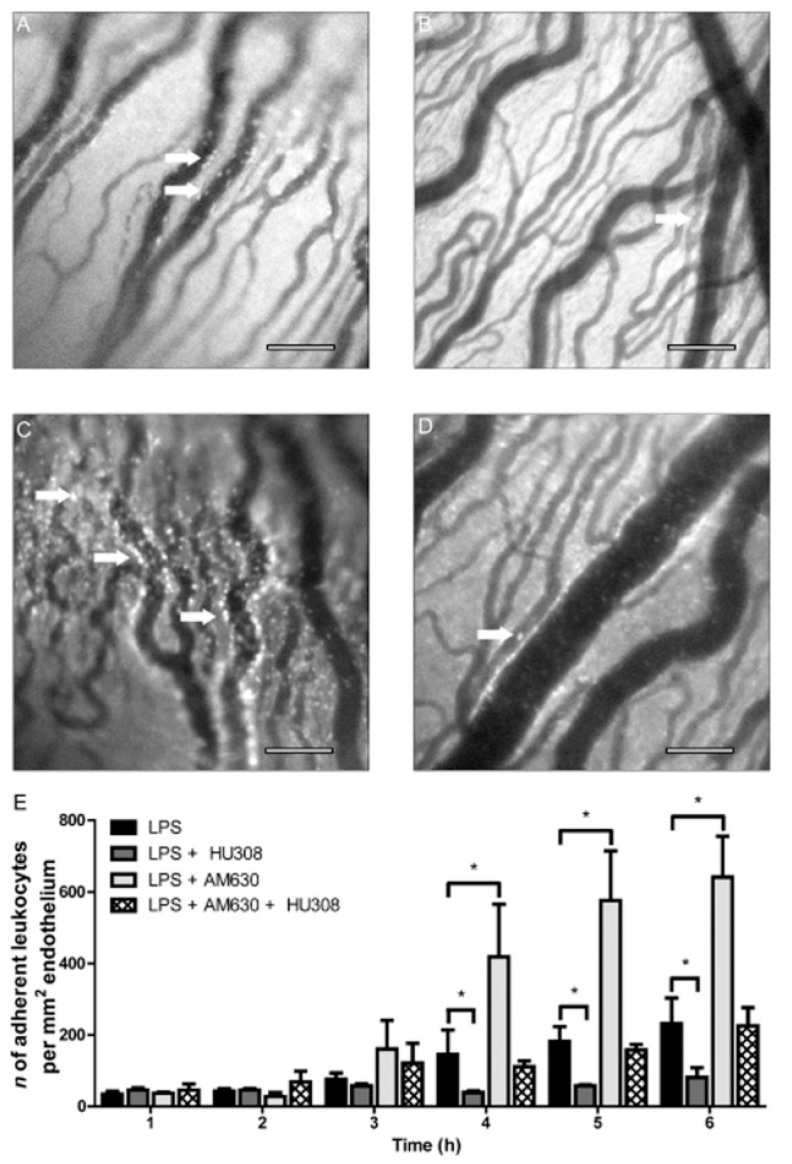
Topical administration of cannabinoid HU308 decreased inflammation in a uveitis mouse model. (**A**–**D**) Representative intravital microscopy images of iridial microcirculation in rat eyes at 6 h after IVT LPS injection in the following groups: (**A**) LPS injection only; (**B**) LPS + CB2 receptor agonist, HU308 (1.5%, topical); (**C**) LPS + CB2 receptor antagonist, AM630 (2.5 mg·kg^−1^, i.v.); and (**D**) LPS + AM630+ HU308. Arrows indicate adherent leukocytes. Scale bar = 100 μm. (**E**) Bar graph representing the time course for the mean number of adherent leukocytes for the above groups including the following: LPS (n = 15); LPS + HU308 (n = 12); AM630 (n = 8); and AM630 + HU308 (n = 7). Values represent the means ± SEM. * *p* < 0.05 compared with the LPS group. Reproduced from Toguri et al. [88].

**Figure 5 pharmaceutics-15-01077-f005:**
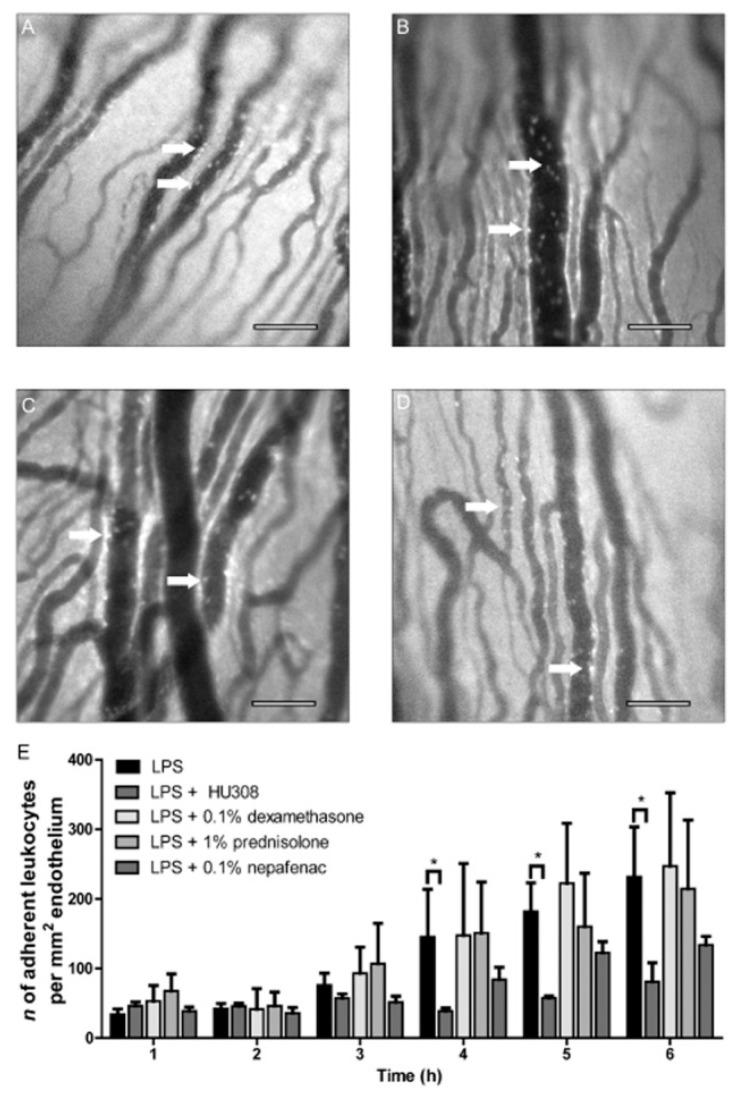
Cannabinoid HU308 outperformed commonly used anti-inflammatory compounds at decreasing inflammation in a uveitis mouse model. (**A**–**D**) Representative intravital microscopy images of iridial microcirculation in rat eyes at 6 h after IVT LPS injection in the following groups: (**A**) LPS injection only (n = 15); (**B**) LPS + dexamethasone (0.1% topical); (**C**) LPS + prednisolone (1% topical); and (**D**) LPS + nepafenac (0.1% topical). Arrows indicate adherent leukocytes. Scale bar = 100 μm. (**E**) Bar graph representing the time course for the mean number of adherent leukocytes in the above treatment groups compared to the LPS + HU308 group (n = 9 for all groups). Values represent the means ± SEM. * *p* < 0.05; compared with the LPS group. Reproduced from Toguri et al. [88].

**Figure 6 pharmaceutics-15-01077-f006:**
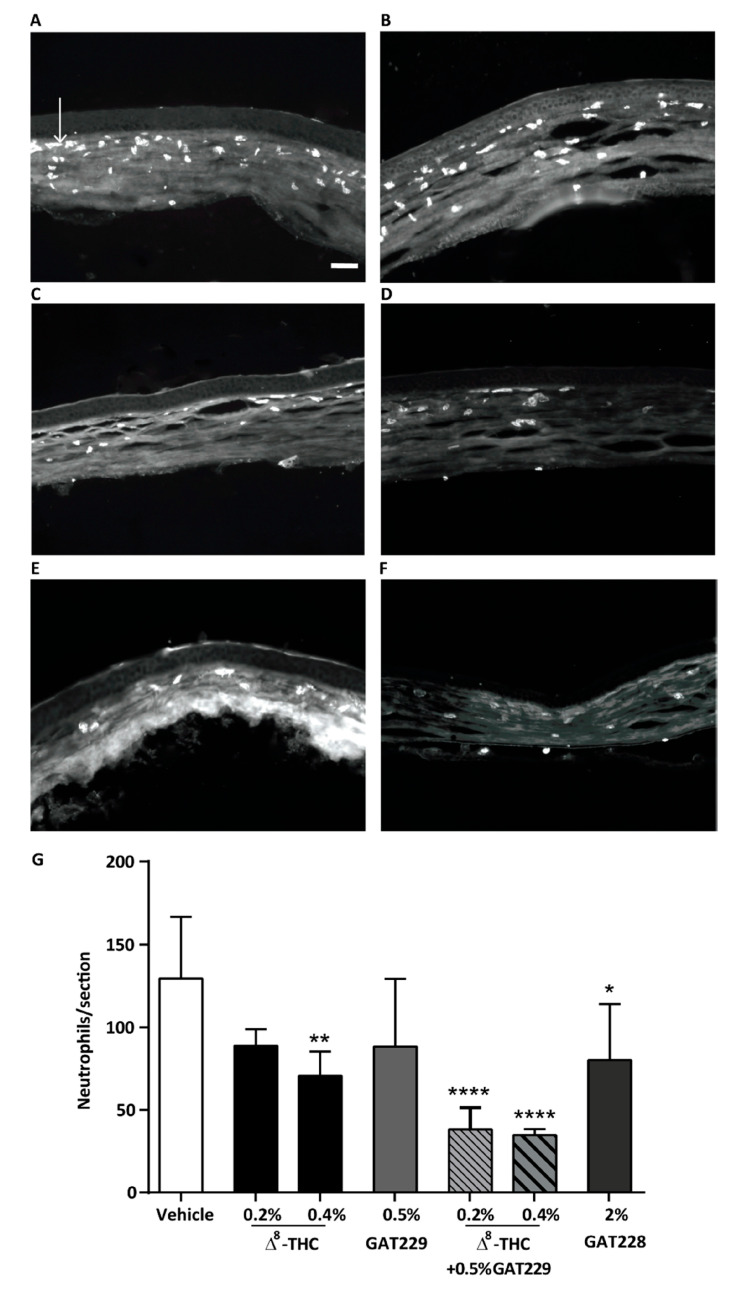
Δ8-THC exerted antinociceptive and anti-inflammatory effects after topical administration in a hyperalgesia mouse model. The figure shows neutrophil expression in cauterized corneas at 6 h post-injury following the topical treatments of drug or vehicles and capsaicin stimulation. Representative images of transverse sections of the central cornea from (**A**) vehicle-treated corneas, (**B**) 0.2% Δ8-THC-treated corneas, (**C**) 0.5% GAT229-treated corneas, (**D**) 2% GAT228-treated corneas, (**E**) 0.2% Δ8-THC + 0.5% GAT229 and (**F**) 0.4% Δ8-THC + 0.5% GAT229. (**G**) Effects of topical treatment of WT-cauterized eyes with (0.2 and 0.4%) Δ8-THC, 0.5% GAT229, 2% GAT228 or 0.2% or 0.4% Δ8-THC + 0.5% GAT229 (n = 4–6 per group) in neutrophil infiltration compared to vehicle-treated eyes (n = 7). GAT211 CB1: allosteric ligand; GAT228 and GAT229: GAT211 enantiomers. Values represent the means ± SD. Arrow in (**A**) points to one of many infiltrating neutrophils. Scale bar: 50 µm. For statistical analysis one-way ANOVA with Dunnett’s post hoc test (compared to the vehicle) was used. * *p* < 0.05, ** *p* < 0.01, **** *p* < 0.0001. Reproduced from Thapa et al. [87].

**Figure 7 pharmaceutics-15-01077-f007:**
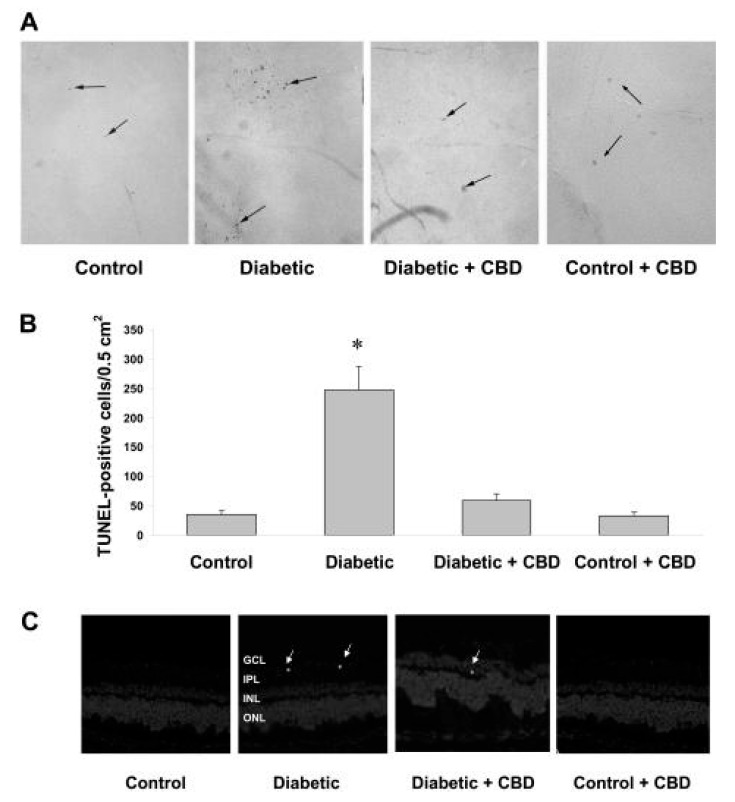
CBD exerted retinal neuroprotective effects in an experimental diabetes rat model. A terminal dUTP nick end labeling (TUNEL) assay was performed after 4 weeks of induced diabetes to detect retinal cell death by using horseradish peroxidase (HRP) detection (TUNEL-HRP). (**A**) Numerous TUNEL HRP-labeled cells (arrows) were detected in whole-mounted retinas from 4-week diabetic rats as compared with untreated controls and the CBD-treated group. (**B**) Total number of TUNEL HRP-positive cells counted in each retina, expressed per 0.5 cm2. The diabetic rats had significantly more TUNEL HRP-positive cells than the controls and the CBD-treated group (* *p* < 0.001; n = 5 to 6). Treatment with CBD (10 mg/kg every 2 days) blocked cell death in the diabetic retinas but did not alter the number of TUNEL+ cells in control rats. (**C**) A representative image shows the TUNEL labeling of frozen eye sections from the diabetic rats (4 weeks) in different retinal layers: ganglion cell layer (GCL), inner plexiform layer (IPL), inner nuclear layer (INL) and outer nuclear layer (ONL). TUNEL+ cells (arrows) were distributed mainly in the inner retinal layers. Original magnifications, ×100 (**A**). Reproduced from El Remessy et al. [116].

## Data Availability

Not applicable.

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
