# Peer review of "Cannabinoid-Based Ocular Therapies and Formulations"

_pharmaceutics, 2023, doi:10.3390/pharmaceutics15041077_

Round 1

Reviewer 1 Report

In the review entitled “Cannabinoid-based ocular therapies and formulations”, the authors describe eye diseases. The work is well organized and interesting for the researchers that read this review.

I suggest to accepted with minor revision

Page 6, line 204, the sentence is incomplete. Please, rewrite it.

Page 8, line 277, the authors wrote “IV”, explain it, please.

Page 10, line 400, the authors wrote “Szcesniak et aI.” Instead of “Szcesniak et al”. Please correct it.

Page 11, line 408, the authors wrote “Kabiri et aI.” Instead of “Kabiri et al.”. Please correct it.

Page 11, line 445, the authors wrote “Sosnik et at.” Instead of “Sosnik et al.”. Please correct it.

Page 18, line 623, the authors wrote “IP administration”, explain it, please.

Page 20, line 715, the authors wrote “El-Galniny et aI.” Instead of “El-Galniny et al.”. Please correct it.

Page 20, line 720, delete “(“, please.

In figure 1, the caption is at the top and bottom of the figure. Please, put it down the figure.

In figure 4 and 5 a,b,c,d the scale bar is not present. Please, introduce it.

It is necessary to improve the resolution of the figures.

Author Response

We would like to express our sincere gratitude to the reviewers for their valuable insight and advice. They certainly helped us improve the quality of our work and provided much appreciated guidance.

The comments have been addressed point by point. The modifications subsequently made, as well as additional minor modifications to maintain the fluency of the text after their incorporation, have been highlighted in the manuscript to facilitate their identification. We sincerely hope this updated version comply with the requested improvement.

In the review entitled “Cannabinoid-based ocular therapies and formulations”, the authors describe eye diseases. The work is well organized and interesting for the researchers that read this review.

I suggest to accepted with minor revision

Page 6, line 204, the sentence is incomplete. Please, rewrite it.

We appreciate the observation and apologize for the mistake. The sentence was rewritten.

Page 8, line 277, the authors wrote “IV”, explain it, please.

We thank the reviewer for the suggestion and apologize for the trouble. “IV” was used standing for “intravenous”, this abbreviation was now incorporated at the first time the word was used in the text (line 143).

Page 10, line 400, the authors wrote “Szcesniak et aI.” Instead of “Szcesniak et al”. Please correct it.

Page 11, line 408, the authors wrote “Kabiri et aI.” Instead of “Kabiri et al.”. Please correct it.

Page 11, line 445, the authors wrote “Sosnik et at.” Instead of “Sosnik et al.”. Please correct it.

We apologize for the repeated mistake and appreciate the correction, the writing was corrected.

Page 18, line 623, the authors wrote “IP administration”, explain it, please.

Likewise, we are grateful for the correction. “IP” was used standing for “intraperitoneal”, which has been clarified in the text in this corrected version.

Page 20, line 715, the authors wrote “El-Galniny et aI.” Instead of “El-Galniny et al.”. Please correct it.

Again we apologize for the mistake and appreciate the observation. The writing (et aI.) was amended to “et al.”

Page 20, line 720, delete “(“, please.

We appreciate the observation and apologize for the mistake, the error was amended.

In figure 1, the caption is at the top and bottom of the figure. Please, put it down the figure.

We are thankful for the observation; the caption was moved to the bottom of the figure.

In figure 4 and 5 a,b,c,d the scale bar is not present. Please, introduce it.

We appreciate the observation. The scale bar was indeed not present within the figure itself form the author source, it was stated in the caption figure (“Scale bar = 100 μm”) and reproduced in this manuscript likewise. To facilitate its localization, it has been highlighted as well.

It is necessary to improve the resolution of the figures.

We are thankful for the observation. The figures, reproduced from the source journals with the maximum resolution available, will be submitted now as independent files instead of only embedded in the manuscript text.

Reviewer 2 Report

Summary:

The present study reflects the authors' efforts to analyse the data of a number of publications related to the role of cannabinoids in the treatment of different types of ocular diseases.

The subject matter is current and interesting, primarily because of the difficulties in the treatment of this diseases.

The authors presented their results in a logical and meaningful manner, using tables and figures for the better interpretation of the data. The conclusions of the work are clear and are well supported by the results. The study will serve as a helpful document in this field of research.

Nevertheless, some minor revisions are recommended before publication.

Observations:

Line 201: Please correct the mistype: “3.1. 3.1.”

Line 202: The first paragraph of this section should be deleted.

Line 304: “w/w” should be “w/v”.

Lines 400, 408, 667, 715: “aI.” should be “al.”

Line 492: “Uveítis” should be “Uveitis”

Lines 530 and 532: “Figura” should be “Figure”

Line 552: “LPS + HU308” is missed from the figure capture

Line 552: “(c) prednisolone (1% topical) and (d) nepafenac (0.1% topical).” should be “(c) LPS + prednisolone (1% topical) and (e) LPS + nepafenac (0.1% topical).”

Line 589: Please correct the mistype: “corneal epithelium and endothelium,…”

Line 596: antagonists” should be “agonists”

Line 739: “arena” should be “area”

Please verify and correct the reference list for consistency, especially regarding to the year of publication.

It is recommended that the manuscript be reviewed by native English speaker.

Summary:

The present study reflects the authors' efforts to analyse the data of a number of publications related to the role of cannabinoids in the treatment of different types of ocular diseases.

The subject matter is current and interesting, primarily because of the difficulties in the treatment of this diseases.

The authors presented their results in a logical and meaningful manner, using tables and figures for the better interpretation of the data. The conclusions of the work are clear and are well supported by the results. The study will serve as a helpful document in this field of research.

Nevertheless, some minor revisions are recommended before publication.

Observations:

Line 201: Please correct the mistype: “3.1. 3.1.”

Line 202: The first paragraph of this section should be deleted.

Line 304: “w/w” should be “w/v”.

Lines 400, 408, 667, 715: “aI.” should be “al.”

Line 492: “Uveítis” should be “Uveitis”

Lines 530 and 532: “Figura” should be “Figure”

Line 552: “LPS + HU308” is missed from the figure capture

Line 552: “(c) prednisolone (1% topical) and (d) nepafenac (0.1% topical).” should be “(c) LPS + prednisolone (1% topical) and (e) LPS + nepafenac (0.1% topical).”

Line 589: Please correct the mistype: “corneal epithelium and endothelium,…”

Line 596: antagonists” should be “agonists”

Line 739: “arena” should be “area”

Please verify and correct the reference list for consistency, especially regarding to the year of publication.

It is recommended that the manuscript be reviewed by native English speaker.

Author Response

We would like to express our sincere gratitude to the reviewers for their valuable insight and advice. They certainly helped us improve the quality of our work and provided much appreciated guidance.

The comments have been addressed point by point. The modifications subsequently made, as well as additional minor modifications to maintain the fluency of the text after their incorporation, have been highlighted in the manuscript to facilitate their identification. We sincerely hope this updated version comply with the requested improvement.

Summary:

The present study reflects the authors' efforts to analyse the data of a number of publications related to the role of cannabinoids in the treatment of different types of ocular diseases.

The subject matter is current and interesting, primarily because of the difficulties in the treatment of this diseases.

The authors presented their results in a logical and meaningful manner, using tables and figures for the better interpretation of the data. The conclusions of the work are clear and are well supported by the results. The study will serve as a helpful document in this field of research.

Nevertheless, some minor revisions are recommended before publication.

 Observations:

Line 201: Please correct the mistype: “3.1. 3.1.”

We apologize for the mistake and appreciate the observation, the mistype was amended.

Line 202: The first paragraph of this section should be deleted.

We are thankful for the correction and apologize for the trouble, the duplicated paragraph was deleted.

Line 304: “w/w” should be “w/v”.

Again, we sincerely appreciate the observation. Actually, the concentration 0.1% w/w is referred to ophthalmic suspensions. Thanks to the reviewer correction and after carefully checking the source data, the text has been modified to clarify the maximum allowed concentrations for ophthalmic administration as follows:

“In specific, polysorbate 80 is approved for ophthalmic application in the form of solution/drops, and emulsions, up to 0.15% and 4% w/v, respectively, while ethanol (alcohol) in solution form is approved for this application up to 0.5 % w/v.”

Lines 400, 408, 667, 715: “aI.” should be “al.”

We apologize for the errors; the text has been amended in the corresponding places.

Line 492: “Uveítis” should be “Uveitis”

We appreciate the correction; the text was subsequently amended.

Lines 530 and 532: “Figura” should be “Figure”

Again we apologize for the mistakes and thank for the correction, the text was modified at both locations accordingly.

Line 552: “LPS + HU308” is missed from the figure capture

We are thankful for the observation. The group “LPS + HU308” has been included in the figure caption in reference to the graphic corresponding to Figure 5E, since no microscopy image representative from this group was included by the authors in the reproduced figure.

Line 552: “(c) prednisolone (1% topical) and (d) nepafenac (0.1% topical).” should be “(c) LPS + prednisolone (1% topical) and (e) LPS + nepafenac (0.1% topical).”

Once again we sincerely appreciate the observation. The caption was modified accordingly.

Line 589: Please correct the mistype: “corneal epithelium and endothelium,…”

We thank the reviewer for the observation. This initial typing is in accordance to the cited work, we are afraid we were not able to identify the corresponding correction. We apologize for any trouble this may cause.

Line 596:antagonists” should be “agonists”

We are grateful for the observation. The intended term was indeed antagonists, but we could understand from this observation that the initial text could be improved for clarity. The phrase was rectified accordingly.

Line 739: “arena” should be “area”

We appreciate the observation. The expression “in this arena” was changed to “in this area”.

Please verify and correct the reference list for consistency, especially regarding to the year of publication.

We are grateful for the recommendation. The reference list was thoroughly revised and corrected.

It is recommended that the manuscript be reviewed by native English speaker.

We appreciate the recommendation. The manuscript text was revised attending to the standard of quality requested.

Reviewer 3 Report

pharmaceutics-2219276

Cannabinoid-based ocular therapies and formulations

The manuscript by Saraiva et al. summarized eye disease conditions potentially subject to treatment with cannabinoids (such as glaucoma, uveitis, diabetic retinopathy, keratitis, and the prevention of Pseudomonas aeruginosa infections) and previous studies that have developed cannabinoid-based formulations for ocular administration. Overall, the manuscript was well prepared. Below are some recommendations to improve the manuscript.

1. In section 2, there is a part introducing different nanosystems. However, this part is superficial and should be expanded. The authors should discuss these nanosystems in detail (such as structure, components, properties, advantages in ocular delivery, etc.). Some information in section 3 can be placed here. It is critical to understand the nanosystems before discussing their applications in the delivery of cannabinoids for ocular therapies.

2. In subsection 3.1 of section 3, the authors should re-arrange the information into two parts: 3.1.1. Therapeutic potential/ effects of cannabinoid on glaucoma and limitations of conventional formulations, and 3.1.2. Recent advances of nanosystems in enhancing therapeutic effects of cannabinoids on glaucoma. Similar to other subsections of section 3.

3. Table 1 only includes some formulations developed for glaucoma and keratitis. It should be expanded. In this table, the authors should summarize all studies discussed in six subsections of section 3.

4. Are there any cannabinoid-based ocular nanoformulations in clinical practice or clinical trials? Please discuss this in sections 4 or 5.

5. There is subsection 2.1 but not 2.2.

6. Line 201: 3.1.3.1 should be 3.1.

7. Typos: “Figura” => “Figure”.

Author Response

We would like to express our sincere gratitude to the reviewers for their valuable insight and advice. They certainly helped us improve the quality of our work and provided much appreciated guidance.

The comments have been addressed point by point. The modifications subsequently made, as well as additional minor modifications to maintain the fluency of the text after their incorporation, have been highlighted in the manuscript to facilitate their identification. We sincerely hope this updated version comply with the requested improvement.

The manuscript by Saraiva et al. summarized eye disease conditions potentially subject to treatment with cannabinoids (such as glaucoma, uveitis, diabetic retinopathy, keratitis, and the prevention of Pseudomonas aeruginosa infections) and previous studies that have developed cannabinoid-based formulations for ocular administration. Overall, the manuscript was well prepared. Below are some recommendations to improve the manuscript.

  1. In section 2, there is a part introducing different nanosystems. However, this part is superficial and should be expanded. The authors should discuss these nanosystems in detail (such as structure, components, properties, advantages in ocular delivery, etc.). Some information in section 3 can be placed here. It is critical to understand the nanosystems before discussing their applications in the delivery of cannabinoids for ocular therapies.

We appreciate the suggestion. Section 2 has been extended to accommodate further discussion on the requested matter.

  1. In subsection 3.1 of section 3, the authors should re-arrange the information into two parts: 3.1.1. Therapeutic potential/ effects of cannabinoid on glaucoma and limitations of conventional formulations, and 3.1.2. Recent advances of nanosystems in enhancing therapeutic effects of cannabinoids on glaucoma. Similar to other subsections of section 3.

We thank the reviewer for the observation and completely agree with the suggestion. Actually, this subsection 3.1 follows the same structure as the rest of subsections in section 3, mainly; i) specifics of the disease; ii) limitations of current treatments; iii) pharmacological potential of cannabinoids based on available data; iv) opportunities through advanced drug delivery. While stablishing additional subsections was initially considered, it was initially discarded due to the short length of the text corresponding to subsections 3.2 to 3.6 and for the sake of consistency through the subsections. However, we could understand now from the reviewer correction that additional subsections were indeed needed in this 3.1 subsection. It has been now re-arranged in the subsections 3.1.1 and 3.1.2 suggested by the reviewer.

  1. Table 1 only includes some formulations developed for glaucoma and keratitis. It should be expanded. In this table, the authors should summarize all studies discussed in six subsections of section 3.

We appreciate the suggestion. Table 1 summarized reported cannabinoid nanoformulations for ocular administration, which up to date have been developed only for glaucoma and keratitis. The table has been extended, and the corresponding heading and caption modified, to accommodate now reported cannabinoid formulations (not only nanotechnology-based) for ocular administration.

  1. Are there any cannabinoid-based ocular nanoformulations in clinical practice or clinical trials? Please discuss this in sections 4 or 5.

We are thankful for the observation. To the best of our knowledge, and based on the available literature and clinical trials information on the clinicaltrials.gov database, currently there are no clinical data available on cannabinoid ocular delivery by means of nanoformulations, hence no comment was initially dedicated to this matter. We understand from the suggestion that some discussion on the issue was convenient. A brief mention and discussion on this matter was included in Section 4.

  1. There is subsection 2.1 but not 2.2.

We appreciate the observation. That is correct, there is a subsection 2.1 (Properties of ocular drug delivery nanosystems), but no additional subsections were considered necessary in Section 2 considering the organization of the information displayed along the text.

  1. Line 201: 3.1.3.1 should be 3.1.

We apologize for the typo and thank the reviewer for the correction, the mistake was amended.

  1. Typos: “Figura” => “Figure”.

We appreciate the correction; the mistake was corrected along the text.

Reviewer 4 Report

Dear Authors,

the paper titled: "Cannabinoid-based ocular therapies and formulations” is a good collection of articles concerning the ocular application of cannabinoid formulations.

Some remarks:

Unfortunately, the various European legislations do not all provide for the use of cannabis for eye diseases.

I would also insert this clarification in the text (introduction) and in support of some reference papers:

11)     Healthcare 2021, 9, 1425. https://doi.org/10.3390/ healthcare9111425

22)     Eur Rev Med Pharmacol Sci 2018; 22 (4): 1161-1167 DOI: 10.26355/eurrev_201802_14405

With regard to ocular nanotechnological formulations, I would suggest to insert some references that can underline how important and studied this route of drug administration.

11)     Pharmaceuticals 2022, 15, 1087. https://doi.org/10.3390/ph15091087

22)     Progress in Retinal and Eye Research 29 (2010) 596- 609, doi: 10.1016/j.preteyeres.2010.08.002.

Author Response

We would like to express our sincere gratitude to the reviewers for their valuable insight and advice. They certainly helped us improve the quality of our work and provided much appreciated guidance.

The comments have been addressed point by point. The modifications subsequently made, as well as additional minor modifications to maintain the fluency of the text after their incorporation, have been highlighted in the manuscript to facilitate their identification. We sincerely hope this updated version comply with the requested improvement.

Dear Authors,

the paper titled: "Cannabinoid-based ocular therapies and formulations” is a good collection of articles concerning the ocular application of cannabinoid formulations.

Some remarks:

Unfortunately, the various European legislations do not all provide for the use of cannabis for eye diseases.

I would also insert this clarification in the text (introduction) and in support of some reference papers:

1)     Healthcare 2021, 9, 1425. https://doi.org/10.3390/healthcare9111425

2)     Eur Rev Med Pharmacol Sci 2018; 22 (4): 1161-1167 DOI: 10.26355/eurrev_201802_14405

We thank the reviewer for the valuable comment. The indicated remark and references were included in the introduction section as follows:

 “Even though many legislations, specifically within the EU, do not currently provide for the use of cannabis for the treatment of eye diseases [11,12], the current climate of renewed interest on cannabis medical use and legalization entice the research in this area.”

With regard to ocular nanotechnological formulations, I would suggest to insert some references that can underline how important and studied this route of drug administration.

11)     Pharmaceuticals 2022, 15, 1087. https://doi.org/10.3390/ph15091087

22)     Progress in Retinal and Eye Research 29 (2010) 596- 609, doi: 10.1016/j.preteyeres.2010.08.002.

We appreciate the pertinent suggestion. The suggested references were included in the manuscript along with the following text now reading in section no. 2:

“Their capacity to protect labile drugs against degradation, and overcome ocular barriers, makes them an especially valuable tool for ocular drug delivery in general, and for the ocular delivery of challenging molecules (i.e. biomacromolecules, poorly water-soluble molecules) in particular [33,34].”

Round 2

Reviewer 3 Report

All the raised concerns were appropriately resolved. The manuscript can be accepted as is.

Author Response

We appreciate the further suggestions to improve the quality and originality of the work.

Figure 1 has been substituted by an original image focused on the effects of cannabinoids at the ocular level.

The “Conclusions” section title has been changed to “Conclusions and future perspectives”, and a final paragraph including a brief summary and the future perspectives that the authors envisage in the field was included.

We sincerely hope the modifications comply with the improvement requested.